# Beyond Blue and Green Spaces: Identifying and Characterizing Restorative Environments on Sichuan Technology and Business University Campus

**DOI:** 10.3390/ijerph192013500

**Published:** 2022-10-19

**Authors:** Yue Du, Zan Zou, Yaodong He, Yongge Zhou, Shixian Luo

**Affiliations:** 1College of Art, Sichuan Technology and Business University, Chengdu 610000, China; 2College of Landscape Architecture, Sichuan Agricultural University, Chengdu 611130, China; 3College of Chemistry, Sichuan University, Chengdu 610000, China; 4Department of Environmental Sciences and Landscape Architecture, Graduate School of Horticulture, Chiba University-Matsudo Campus, Chiba 271-8510, Japan

**Keywords:** restorative environments, campus, participatory smartphone photography

## Abstract

Undergraduates commonly suffer from stress and anxiety; therefore, it is imperative to find restorative places on campus. Although blue and green spaces are good for recovery and stress relief, previous studies have failed to determine other types of restorative spaces on campuses. Using a bottom-up participatory smartphone photo survey, this study recruited a sample of 243 students from Sichuan Technology and Business University in China, and the results were as follows: (1) potential restorative spaces on campus were grouped into five categories: green, blue, gray, living, and study space; (2) no significant differences were found in the assessment of the five restorative spaces, all of which showed positive effects; (3) the five restorative spaces were linked with four restorative characteristics in different ways, with green, blue, gray, and living space showing the “being away” characteristic (refuges from the hassles of everyday life, indicate geographical or psychological distance), and gray and study spaces showing the “fascination” characteristic (effortless attention); (4) visit duration played an important role in the environment’s potential to promote recovery. A shorter visit duration owing to a lack of infrastructure and interest points may contribute to reduced benefits. This study has important implications for the design and management of restorative environments on college campuses.

## 1. Introduction

Many undergraduates suffer from negative emotions, such as stress, depression, and anxiety, as a result of dealing with academic, interpersonal, economic, and cultural challenges [1,2,3]. About half of college students have moderate levels of stress-related mental health problems, including anxiety and depression [4]. According to the latest report on national mental health development in China (2019–2020) [5], approximately 18.5% of college students suffer from depression, and their mental health problems are increasing. According to the American College Health Association, overall anxiety among college students has increased by 14.2% since 2000 [6]. The university campus is a place where students experience repeated scene patterns and engage in socialization. The campus environment has the potential to become key in improving the physical and mental health of students [7]. In China, students spend most of their time on campus because of the boarding system. Because of the travel restrictions set in place during the COVID-19 pandemic, students spend even more time on campus. Therefore, it is necessary and urgent to investigate restorative sites on campus [8].

There are two theoretical approaches to the study of restorative environments: stress recovery theory (SRT) and attention restoration theory (ART) [9,10], which argue that humans have evolved over a long period in the natural environment; thus, people are, to some extent, physically and psychologically more adapted to nature than to urban environments [11]. ART conveys that humans tend to focus on natural content (e.g., vegetation and water) and environments conducive to survival [12,13]. Both theories agree that attention restoration and mood regulation are effective in natural environments. Therefore, in recent years, many studies have focused on green and blue spaces related to the natural environment of university campuses, which have been proven to have a positive impact on the mental health of college students. Tree-lined walkways on campus can enhance students’ perception of resilience [14], and the green space gives students the opportunity to easily immerse themselves in soft fascination with nature during recess, freeing them from the stress of work and study [15]. Green spaces have the ability of helping students relax [16]. The quality and quantity of green spaces can be considered in campus construction to reduce student stress [16]. Attention restoration is also associated with nature [12,17]. Blue spaces (a natural environment dominated by water bodies, which correspond to green spaces, e.g., wetlands, rivers, lakes, and ponds) are considered to have the best restoration potential among outdoor campus spaces [8,18]. In a campus-centered landscape study, Yang et al. found a positive correlation between depression prevention and increased water coverage [19]. Natural elements can influence the resilience of an indoor space. Some researchers believe that viewing real or simulated natural landscapes indoors has a significant impact on resilience. Natural scenery in the classroom window can improve students’ psychological health. A classroom with windows facing green spaces can help relieve stress and mental fatigue in students [20]. A natural mural, particularly one with a waterscape, has high restoration potential in indoor campus environments [21].

However, most of these studies are based on the researcher’s top-down assumption, which means that they were conducted in a proven restorative campus environment (green and/or blue spaces) instead of a bottom-up investigation. Furthermore, during those studies, the types of activities and even the sources of restorative experiences of students on campus were often not limited to green and blue spaces; as a result, the restorative value and potential of other spaces/environments might have been overlooked. Therefore, this study was conducted through a participatory smartphone photography survey to investigate the restorative potential of different spaces on campus from the students’ first-hand perspective [22]. This method has been proven to be effective in many previous studies and is widely used to capture the interaction between people and the environment. For example, Jiang et al. conducted environmental psychology research on sweatshops by asking factory workers to film outdoor factory scenes themselves [23]. Taking advantage of the high visualization and real-time nature of visitor-employed photography (VEP), MacKay et al. captured images uploaded by tourists using the themes and visual effects generated by visitors and their expressions to explore the practicality of tourist photography as a method of obtaining images of destinations [24]. A study by Kyoto University used VEP technology to collect photographic indicators at different scale levels to examine how landscape elements and spatial configurations affect visitors’ on-site experiences on the pilgrimage route [25].

Previous studies have indicated that many types of sites have restorative potential, such as forests, gardens, parks, green roofs, libraries, playgrounds, museums, canteens, zoos, and classrooms [26,27,28,29,30,31,32,33,34,35]. However, very little is known about other site types on campus that have the potential to be beneficial to students’ mental health, except for the commonly considered green and blue spaces. Therefore, using student participatory surveys on a wider campus, this study aimed to delve deeper into a wider range of places on campus other than green and blue spaces.

In addition, previous studies inspired the design of this study. First, Zhang pointed out that certain experiences can make a place restorative [36]. Therefore, in this study, participants were asked about their main activities and behaviors in these spaces to discuss the connection between activities and restoration. Second, Xie found that utilization patterns (frequency and duration of visits) may affect restorative benefits, which may provide a reference for future health interventions targeting college students [37]. Therefore, data on the utilization of these spatial patterns among college students were collected in the survey. Third, there are limitations in mining data solely from volunteer photos, making it difficult for researchers to infer information from the photos, such as why the photos were taken (i.e., the motivation for using the space). Therefore, referring to previous studies [38,39], photo and text data were combined to analyze the motivation of volunteers to use these spaces. Finally, according to environmental psychology, ART is discussed in the context of the restorative characteristics of an environment [40].

In summary, the five research objectives were as follows:Besides blue and green spaces, what are the potential restorative spaces on campus?What are the main activities of students in various restorative spaces?How do utilization patterns affect the restorative potential of these spaces?What is the motivation of the students for using each restorative space?What are the restorative environment characteristics of each restorative space?

## 2. Materials and Methods

### 2.1. Study Area

Sichuan Technology and Business University of China was selected as the study area. Located in Chengdu, China, the chosen university covers an area of 2076.15 mu (about 139.1 ha) and has 13,219 students. Pictures of the school are shown in Figure 1.

### 2.2. Participants

A total of 250 undergraduate students were recruited to participate in the “Fundamentals of Environmental Design” course, which aims to provide students with a hands-on understanding of the methods and elements of environmental design. Participation in the survey was an integral part of the course, and students received credit for the course upon completion of the survey. A total of 250 students participated in the course; however, only 243 students participated in the experiment, after excluding incomplete survey samples (those who did not complete the questionnaire). While 243 participants may be considered a small sample, the number of participants in previous studies was typically 29 to 95 [23,25,38,39]; consequently, 243 samples were considered acceptable for comparison. The sample consisted of 120 male and 123 female students (with a male-to-female ratio close to 1:1), most of whom lived in urban areas for a long time (85.3%) and ranged in age from 17 to 24 years. The study was conducted in accordance with the guidelines of the Declaration of Helsinki and approved by the Academic Council of the university. However, the study was noninvasive and did not investigate human or physiological data; therefore, the academic board indicated that there was no need to submit material for ethical review.

### 2.3. Experimental Design and Investigation Procedure

Participants were asked to take a photo inside the school and complete a questionnaire based on it, limited to the daytime for clearer photos. The photographic survey was conducted from March 20 to April 1. The photo was taken based on the following hypothesis: ‘According to your own experience, if you feel mental fatigue and stress due to various reasons (such as study, examination) during school, which place will you usually choose for recovery?’.

The questionnaire consisted of four parts. The first part asked about basic information (name, sex, and age). The second part aimed to understand the type of activity of the participants in the venues shown in the photos, as well as their utilization patterns. Utilization patterns were divided into two questions: usage duration (‘What is the average time spent here per visit?’ where 1 = less than 30 min, 2 = 30–60 min, 3 = 1–2 h, and 4 = more than 2 h) and frequency of visits (“How often have you visited the site so far this semester?” where 1 = Hardly visited, 2 = Occasionally, 3 = Frequently, and 4 = Almost daily). The third part measured the restorative potential of the site and the restorative environmental characteristics. First, as Jiang asserted [23], we asked participants for their perceived restorative assessment of these sites using a simple question: ‘Based on your own actual visit experience, to what extent do you think this place meets your needs for stress relief and spiritual recovery?’ On a seven-point Likert scale, 1 means not at all and 7 means extremely good. Second, the recovery component scale was used to measure the environmental characteristics of the restorative environment [41], including 12 items to measure the four restorative characteristics of being away, fascination, coherence, and compatibility, and all items were measured by a 7-point Likert scale from “1 = not at all” to “7 = very much”. The fourth part of the questionnaire asked participants to use simple sentences to describe the site, such as the reasons for visiting, experiences, and feelings. This was an open-ended question to examine students’ motivations for visiting various potential restorative spaces, and multiple responses were received.

### 2.4. Data Analysis

Data from 243 questionnaires were collected and used for statistical analysis. Microsoft Excel was used to collect the experimental data. First, variance analysis was performed on the recovery scores of the five spaces, and then we continued to perform a post hoc test (LSD). The chi-square test was then used to check for differences in activity across the spaces. Hayes (2017) PROCESS Macro Version 3.4 was used to conduct a mediation analysis to explore the impact of utilization patterns and restoration potential [42]. Finally, linear regression analyses were performed to explore the environmental features of the five spaces that significantly influenced the potential for restoration. All statistical analyses were performed using the Statistical Package for Social Sciences (SPSS; Version 20.0; SPSS Inc., Chicago, IL, USA).

## 3. Results

### 3.1. Potential Restorative Space on Campus

To explore the potential restorative space on campus, we first analyzed the spatial types based on 243 collected photo samples. According to the photo results, the main spaces involved were the campus sports field (12.76%), Octagon Pavilion pond (11.93%), canteen (9.88%), dormitory (9.05%), Swan Lake (9.47%), library (5.76%), academic building corridor (4.94%), and groves (4.12%). After sorting, discussing and referring to similar studies [8,21,43], the spaces involved in these photos were divided into five main types (Table 1): green, blue, gray, living, and study spaces. Green spaces, such as small gardens, trees, and lawns, are spaces on campus in which greenery is the main environmental element. Blue spaces are mainly water environments and water-bearing areas [44], such as Swan Lake and Octagonal Pavilion pond. Gray spaces are mainly for pavements where sports activities and commuting are carried out, such as football fields, basketball courts, and campus roads. Living spaces are mainly related to students’ daily life activities, where they can sleep, eat, shop, and have fun, such as in dormitories, canteens, and school supermarkets. The study space is defined as the space directly related to studying, such as a library, classroom, or study room.

### 3.2. Potential Restoration Outcomes for Five Restorative Spaces

The third part of the questionnaire asked the students about their assessment of the restorative potential of the spaces in the photographs. According to the data summary statistics (Table 2), the study space had the lowest score (4.19 ± 1.11) and was considered the space with the lowest recovery potential, followed by green space (4.50 ± 1.02), blue space (4.61 ± 1.10), and living space (4.73 ± 1.44). Gray space had the highest recovery score (4.79 ± 1.49). However, the restorative scores of the five spaces were all greater than 4, indicating that most participants rated the restorative experience of the five spaces positively.

A variability analysis was then executed on the restorability scores of the five spaces, followed by a post hoc test (LSD). According to the results (Table 3), no significant difference was found in the perceived recovery scores of the five spaces; that is, the recovery potential of the five spaces was considered to be the same, which is inconsistent with the conclusions in previous literature [8,45]. Previous studies have suggested that natural spaces (green and blue spaces) are the most restorative spaces, which may be related to the study sample and design. We have outlined the reasons for these differences in the discussion section.

### 3.3. Activity Statistics for the Five Restorative Spaces

Table 4 shows the statistical analysis of the types of activities performed in each space. In addition, the option “other” was added to the questionnaire if the researchers missed other potential activities. However, there was no response to this option. As a result, it was removed, and the current results can effectively represent all potential activity types. Furthermore, a direct comparison of frequency sizes is not reasonable because of the inconsistent number of samples in each space. Therefore, in addition to statistical frequency, we introduce the concept of activity index, which is calculated as follows:H_i_ = F_i_/P_i_, i = 1, 2, 3, 4, 5.
where i represents the space type, with a total of five types of spaces.

F is the overall activity frequency of a space;

P is the sample number of respondents in this space;

H is activity index. The larger the index, the more activities selected by a single sample: that is, a single respondent tends to carry out more activities in this space.

The results showed that the activity index of blue space was the highest (H = 3.47), indicating that the respondents tended to carry out more activities in blue spaces. The most frequent activities in blue spaces were landscape viewing (42), breathing fresh air (31), resting (31), and chatting (30). The green space activity index was H = 3.12, and the most selected activities were walking (22), breathing fresh air (18), and enjoying scenery (16). Eating (40), resting (30), and chatting (29) were the most preferred activities in the living space (H = 3.08). The gray space activity index was H = 2.89, and the most selected activities were walking (48), sports (45), and chatting (29). Finally, the activity index of the study space had the lowest value of H = 2.55, and the most selected activities in this space were rest (17), reading (15), and meditation (13). Thus, both chatting and resting are important in most restorative spaces, and fresh air and views are the main activities that promote recovery in outdoor spaces (blue, green, and gray spaces). In addition, the activity index showed that green, blue, and living spaces had higher activity tendencies, and students chose to engage in more activities in these restorative spaces. According to the results of the chi-square test, all activities except meditation showed significant differences among the five spaces. In other words, the respondents had the same tendency to choose meditation in the five spaces, whereas other activities had different tendencies.

### 3.4. Five Patterns of Recovery Space Utilization

Figure 2 shows that the living space had the highest use frequency (3.56 ± 0.68) and the longest duration time (2.40 ± 1.27). Blue space had the lowest use frequency (2.18 ± 0.76) and the shortest duration time (1.95 ± 0.85). The frequency of using green space was medium (2.71 ± 0.84), while the use duration of green space was the shortest (1.85 ± 0.82). In general, living spaces have the longest use duration and the highest use frequency, while blue and green spaces have shorter use durations, and blue space has the lowest use frequency.

Subsequently, a mediation analysis (for all samples) was performed to explore the potential mechanisms of influence between the variables (Figure 3). Previous studies have shown that positive effects are positively correlated with the frequency and duration of visits [46,47,48]. Therefore, we introduced the duration and frequency of visits into the mediation model. The independent variable X is “visit duration,” the dependent variable Y is “recovery potential,” and the mediating variable M is “visit frequency”. The results of this mediation model showed data for all respondents, indicating the relationship between overall spatial volume use patterns and respondents’ perceived restorative benefits. As shown in Figure 3, the effect of frequency and recovery potential (SB = 0.0554, *p* = 0.3658) was not significant; that is, there was no direct correlation between the frequency of space use and restorative potential, and the frequency did not affect recovery. In addition, the frequency variable had no mediating effect in the mediation model (*p* > 0.05), indicating that the frequency of visits did not affect the relationship between utilization duration and recovery potential. Duration, on the other hand, was related to recovery potential (*p* < 0.001); therefore, the longer the duration of each visit, the higher the degree of recovery. Overall, longer visits to restorative spaces on campus, even if the frequency of visits is low, can have a significant positive effect, which is consistent with previous studies [37]. The results indicate that even if students use the spaces more times per day, this does not necessarily lead to a higher recovery. However, longer stays and interactions promoted stronger recovery, which has important implications for the design of campus restorative spaces.

### 3.5. Five Motivations for Using Restorative Spaces

Based on the fourth part of the questionnaire, Table 5 shows students’ descriptions of the photos taken, such as the reasons for the visit, experiences, and feelings. According to the statistical analysis results, students mainly focused on five main motivations in these five spaces: “scenery,” “atmosphere,” “interactive experience,” “outdoor activities,” and “daily life needs.” Specifically, the most common reasons for using green space were “to enjoy natural scenery” (22), “to relax” (19), and “to be quiet” (16). In blue space, “water views” (36), “quiet atmosphere” (32), and “beautiful and romantic feeling” (22) were the main motivations. The main motivations for using gray space are “exercise” (29), “recreation” (26), and “empty and free experience” (22). The main reasons for the use of living space were “lively and pleasant atmosphere” (22), “to meet the needs of daily life” (21), and “for rest and relaxation” (18). Finally, the main reasons for using the study space were “quiet atmosphere” (21), “recess” (19), and “study and practice” (9). Among these, enjoying natural scenery, watching the water scene, exercising, experiencing a lively and pleasant atmosphere, and experiencing a quiet atmosphere were the most common motivations for the five types of space choices.

### 3.6. Restorative Environmental Characteristics of Five Restorative Spaces

A linear regression analysis was performed to explore the environmental features that significantly influenced restorative potential in the five spaces. Residuals, ANOVA, and multi-collinearity tests were performed. First, the Kolmogorov–Smirnov test was used to show that the residuals follow a normal distribution. The results of the ANOVA in the regression analysis showed that there was a significant relationship between the recovery potential of the five spaces and the four recovery characteristics: green space (F = 6.136, *p* = 0.001), blue space (F = 10.544, *p* < 0.001), gray space (F = 19.309, *p* < 0.001), living space (F = 15.932, *p* < 0.001), and study space (F = 4.177, *p* = 0.01). According to previous studies (tolerance value <0.2 or VIF > 10, indicating a problem) [49,50], no multicollinearity problems were found in the current model (lowest tolerance = 0.278, highest VIF = 3.832); therefore, our model was considered acceptable.

According to the regression model results (Table 6), in green space (β = 0.412, *p* = 0.028), blue space (β = 0.436, *p* = 0.008), gray space (β = 0.279, *p* = 0.040), and living space (β = 0.393, *p* = 0.028), being away was the main feature of a restorative environment. Fascination was characteristic of gray space (β = 0.393, *p* = 0.003) and study space (β = 0.529, *p* = 0.032). The results are discussed in the following section.

## 4. Discussion

### 4.1. Potential Types of Restorative Spaces on Campuses

This study evaluated the potential restorative spaces in a university campus environment using a participatory smartphone photography survey. At present, most studies are based on a researcher’s own predetermined hypothesis for verification, which is a top-down approach that may result in biased results. In some cases, the real needs of the experimental subjects (e.g., students and groups of young people) may be ignored. In addition, most studies have asked about the benefits of outdoor environments (mostly natural environments), ignoring the living and learning environments in which students spend most of their time.

To fill this research gap, this study designed a bottom-up participatory survey, recruited 243 students from a university in China, and searched for potential restorative places on the school campus. The main findings were that in addition to blue and green spaces, gray, living, and study spaces were all restorative. According to previous studies, green space can reduce mental stress and relieve anxiety [14,15,16], and blue space is believed to contribute to recovery from mental illness and mental fatigue [19,20]. Overall, studies have shown that greater exposure to nature positively affects attentional recovery and stress reduction.

Gray space can also contribute to attention recovery, showing positive psychological recovery potential [43,51]. Studies have shown that spending 30 min walking and meditating in a square can significantly improve mood and concentration [52]. In addition, regular physical activities performed on campus may help relieve stress and restore the spirit [53], and engaging in leisurely physical activities has been described as a method to cope with daily stressors [54]. Therefore, open gray spaces on campus, such as commuting paths, rooftops, and exercise spaces, may provide different levels of restoration.

The function of a space and its physical characteristics such as living and study spaces, are also important aspects to consider. Hartig et al. proposed a socio-ecological model of stress and recovery based on the residential environment, emphasizing the potential role of housing and the influence of social roles on achieving high-quality recovery [17]. Home is the center of everyday life and was found to be a favorite place in previous related studies [55,56]. Previous studies have suggested that individuals are more likely to develop regional attachment and place memory for frequently visited places, which helps transform everyday environments into restorative spaces [43]. Dining halls and supermarkets are the most commonly used places for college students, leading to the restorative potential of these living spaces. In addition, the dormitory window view is related to quality of life. Tennessen and Cimprich found that students who can see nature through dormitory windows have better attention performance than those with architectural views [57]. Therefore, living spaces in college campuses are believed to provide restorative experiences.

In the study space, focused or prolonged use of directed attention will reduce a person’s ability to suppress distraction and stay focused; however, this does not mean that the person does not enjoy the task at hand [10]. The findings regarding study spaces were surprising, as student stress is usually attributed to these environments, but 31 participants photographed these spaces (library and classroom). Although a previous study discussed the psychological benefits of study spaces such as classrooms, it still focused on observable natural elements (e.g., views outside the window and ornamental plants) [58,59]. There has been little discussion on the restorative effects of these spaces, such as classrooms and libraries. In contrast to previous studies, our work focuses on how the study space guides recovery. The recovery potential of the study space will be analyzed and discussed in depth in the future.

### 4.2. Activities, Utilization Patterns, and Visiting Motivation

In contrast to the majority of psychological recovery literature, which emphasizes natural green and blue environments over indoor environments [21,43], in campus environments, emotional recovery and stress reduction do not need to be completely related to nature, and any environment with restorative quality may be a restorative environment. According to the mediation analysis results of utilization patterns, usage time played an important role in restoration. Therefore, although the activity index of green and blue spaces is higher, they still have equivalent recovery ability to that of other spaces, which may be related to the shorter usage time of the two spaces (GS = 1.95 ± 0.85, BS = 2.71 ± 0.84). For both green and blue spaces, participants reported more types of activities, such as walking, meditating, breathing fresh air, and viewing landscapes. However, “outdoor activities” and “sports” were not the main motivations for their visits. Therefore, it is reasonable to speculate that green and blue spaces provide areas for potential outdoor activities for college students. However, unlike the blue and green spaces in urban parks, the lack of infrastructure, such as seating and fitness equipment on the college campuses surveyed (Figure 1), resulted in shorter usage times.

Recovery always takes place in the context of activities that involve some form and degree of participation in the socio-physical environment [60], such as walking, sports, chatting, scenery viewing, and meditation. Walking (48) and exercise (45) have been mentioned many times as activity types in gray space, and numerous studies have proven that exercise is a positive experience with anti-depression and anti-anxiety effects that improve accumulated acute emotions [61,62]. It has been speculated that for students with long periods of mental work and a sedentary lifestyle, physical exercise is a significant stress release compared to the healing of a gentle outdoor landscape. In this study, the activity index of open gray space was low (H = 2.89), but the duration of use was high (GrS = 2.27 ± 0.93), indicating that a small number of activity types and long-term use can also result in a better restorative experience. While ordinary natural environments evoke soft fascination, environments widely classified as areas of sports/recreation are more likely to evoke strong fascination; that is, the mind has high levels of fascination [63]. Therefore, holding sporting and entertainment activities in gray spaces, such as competitions, music festivals, flea markets, and club activities, etc., may result in better restorative experiences for students.

Living space was used the most frequently (3.56 ± 0.68) and for the longest time (2.40 ± 1.27). Here, “lively and pleasant atmosphere” (22), “meeting the needs of daily life” (21), “rest and relaxation” (18), and other keywords were regarded as the motivation for the use of restorative space, and communication and sense of belonging were particularly important. This result is consistent with the findings of Staats et al. [64]. The key words “atmosphere” and “quiet” are often related to the use of spaces that promote solitude, such as green, blue, and study space. On the other hand, “lively” appeared more frequently in keywords related to life and gray space. It is reasonable to speculate that the effectiveness of communication and companionship can increase the recovery score of life and gray spaces, which tends to facilitate more interaction.

H = 2.55 was the lowest activity index in the study space, and rest (17), reading (15), and meditation (13) were the top three activity types, which were consistent with the main reasons for using study space: “quiet atmosphere” (21) and “recess” (19). A quiet environment is a key component of the restorative experience. The presence of many other people is seen as a barrier to recovery, and there should be a greater focus on positive solitary experiences and self-recovery, consistent with research by Subiza-Perez M [43]. Solitude is two-sided and seems to be related to motivation and preferences [65]. Regarding the study space, solitude may be related to stress and negative emotions; however, for students who prefer to be alone, solitude can relieve stress, which may be the reason why 31 participants chose study spaces (classroom, library).

### 4.3. The Environmental Characteristics of Campus Restorative Space

According to the regression analysis results, among green space (β = 0.412, *p* = 0.028), blue space (β = 0.436, *p* = 0.008), gray space (β = 0.279, *p* = 0.040), and living space (β = 0.393, *p* = 0.007), “being away” is the main feature of the restorative environment. “Being away” can indicate geographical or psychological distance. In this study, the expressions “here can let me temporarily escape from study and daily chores,” “the time here so I can get a good relax in fatigue stress,” and “here let me experience is different from usual environment” were used to describe and evaluate how the escape from learning or social pressure helped with adjustment and adaptation to the stress and stimulation of daily life. For natural environment spaces, such as green and blue spaces, being near nature is generally considered to result in a higher perceptual experience, which is attributed to the quality of the natural environment in some studies. People who have full access to nature are reported to have higher recovery [66,67,68]. In terms of life and gray space, students escape from stressors by returning to the dormitory or engaging in sports. As a key word for living space, a sense of belonging emphasizes psychological return, physiological rest, and recovery. The gray space, on the other hand, is used for exercise and to achieve restoration, differentiating it from a sedentary study state.

Fascination was a characteristic of both restorative environments in gray space (β = 0.393, *p* = 0.003) and study space (β = 0.529, *p* = 0.032). Fascination is referred to as effortless attention, and the expressions used to describe it are as follows: “the environment here is fascinating,” “there are a lot of interesting things that attract my attention,” and “it makes me feel pleasant and calm.” Playing and viewing sports in gray space can be regarded as a reflection of fascination. Watching a competitive sporting event is likely to generate intense fascination [21] that leaves little room for other thoughts. Interestingly, learning requires effortful concentration, yet study space fascination has been reported to be a feature of a significantly restorative environment. In the data collected in this study, it was found that students who took pictures of the study space mostly took pictures of the library (14) and the corridor platform of the teaching building (11), while fewer students only took pictures of the classroom (2). In terms of motivation, it was described as follows: “There is a quiet atmosphere ” (21), “Here, I can take a class break” (19), and “I can learn and read instead.” The activity types were mainly rest (17), read (15), and meditation (13), which indicates that the fascination feature was more prominent in quiet environments where there were learning gaps and an interest in reading. “Soft fascination” is referred to as a moderately charming experience of aesthetically pleasurable stimuli [12], which not only requires little effort but also leaves mental space for reflection. Spontaneous thought processes, including reflecting, daydreaming, planning, and other forms of internally generated thinking, are states similar to soft fascination [69]. A quiet environment is conducive to reflection, where reading and resting can engage the mind without filling it up. Therefore, one can also exhibit fascination in study spaces.

### 4.4. Research Significance and Contribution

This research is of great value for the development of restorative campus environments. While most research on restorative environments is limited to natural environments (such as green and blue spaces), this study focuses on all potential restorative environments. A bottom-up participatory smartphone photo survey (PSPS) method was used to obtain information on the types of restorative spaces, utilization patterns, visiting motivations, and restorative characteristics of university students. The results indicate that gray, living, and study spaces all have restorative value, which may be useful in improving the mental health status of college students. This finding is particularly important. Most university students in the world face anxiety and pressure; campus administrators can not only improve the quality of the natural environment, but also focus on potential restorative spaces. Holding regular sporting events and outdoor club activities may relieve stress and restore spirit; living spaces that are used for the longest time and with the highest frequency should be valued. Daily environments such as dormitories, canteens, and supermarkets are more likely to produce regional attachment and place memory, which are important for achieving high-quality recovery.

Further research can be conducted on the orientation of the dormitory structure, interior decoration, window views, dining environment of the canteen, and shopping environment of the supermarkets to promote the improvement in the quality of students’ campus life; the study space is the focus of the school, and the quiet environment is a key component of restorative experience. Through high-quality rest, reading, meditation, and other activities, students can easily reach a soft fascination state.

The research method used in this study, the PSPS, is worth popularizing. It is derived from traditional participatory photo mapping (PPM) developed by Dennis Jr., Gaulocher, and Brown [70]. Perceived as simple, user-friendly, and democratic, it can be used to analyze the advantages and disadvantages of the environment from the user’s perspective, as well as other environments in the city.

### 4.5. Limitations and Future Research

Although we had a large sample size, the results of this study are limited. First, all the students who participated in the survey were from the same major, which may affect the generalizability of our findings. Second, geographic information regarding the photographs was not recorded, and it was impossible to know the specific locations where the photographs were taken, which may have provided other potentially valuable insights. Third, all the pictures were captured during the day. Future studies should explore whether the spatial resilience potential varies over different time periods.

## 5. Conclusions

Although blue and green spaces have already been proven to be restorative in previous research [8,14,15,16,17,18], the results of this study indicate that gray, living, and study spaces on campus are also restorative. To be restorative, the environment does not need to be fully associated with nature. For students who spend a lot of time in a sedentary environment, physical activity might be a more significant stress reliever than an underwhelming outdoor landscape. The effects of solitude are twofold: solitude is more restorative in green, blue, and study spaces, while effective communication and companionship in life and gray spaces can increase the restorative potential. Another novel finding is that there is a significant relationship between the five spatial restorative potentials and four restorative traits. Both study and gray spaces showed fascination traits, with the difference being that students in gray space preferred an atmosphere of sports and passion (intense fascination). Watching and being watched can result in a restorative experience. The students in the study space preferred a quieter atmosphere (soft fascination), and recess and reading interesting books can help students relieve stress quickly during study breaks. Once again, it was proven that the duration of time spent in the environment plays an important role in recovery, and a short duration due to a lack of infrastructure and interest points may contribute to a reduction in recovery. This study also provides ideas for university campus design and later-stage management.

## Figures and Tables

**Figure 1 ijerph-19-13500-f001:**
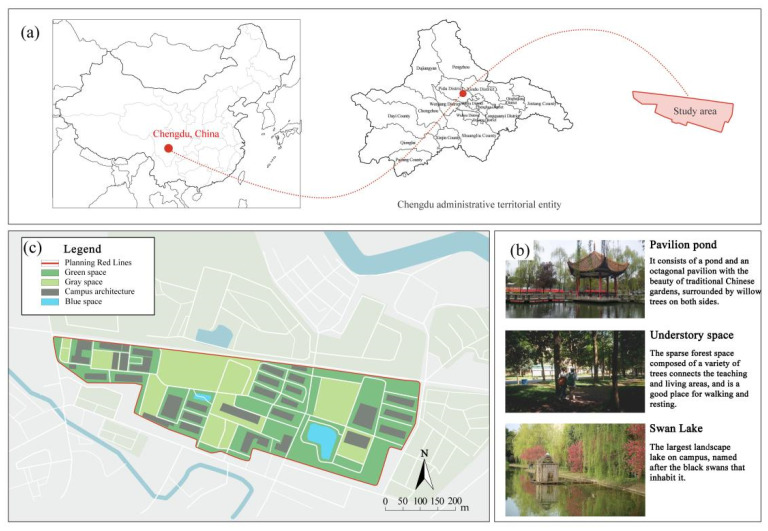
Study area: (**a**) Location of the study site; (**b**) Photos of the campus environment; (**c**) Plan of the study area.

**Figure 2 ijerph-19-13500-f002:**
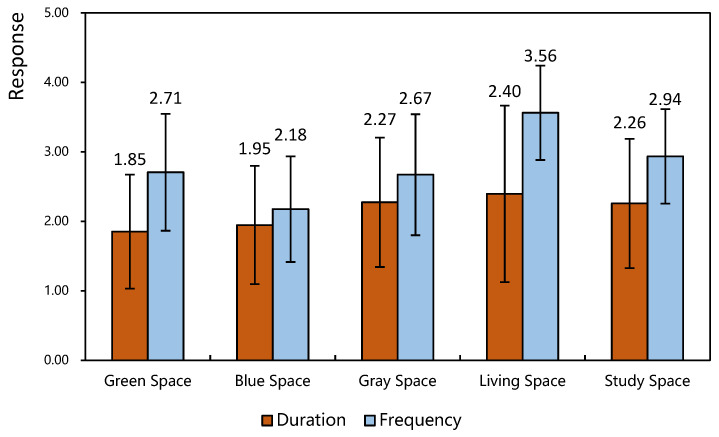
Utilization patterns for five restorative spaces.

**Figure 3 ijerph-19-13500-f003:**
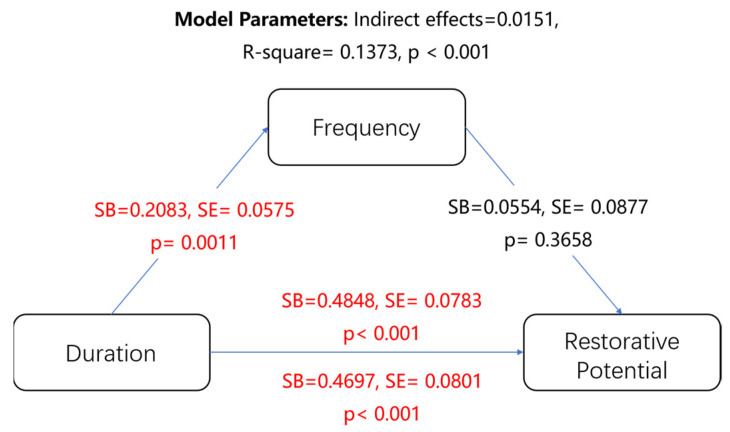
The mediation model between utilization patterns and restoration potential. Note: SB, standardized βs; SE, standard error; red figure, significant path. The upper value represents the total effects; the lower value represents the direct effects.

**Table 1 ijerph-19-13500-t001:** Results of campus restorative spaces classification based on photos.

Category	Image Samples			Description
Green Space	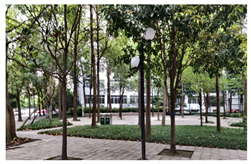	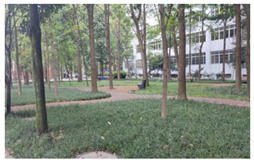	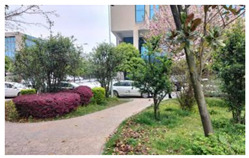	A landscape space composed of plants
Blue Space	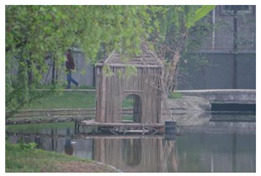	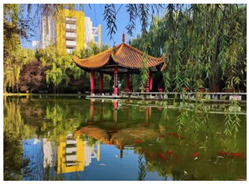	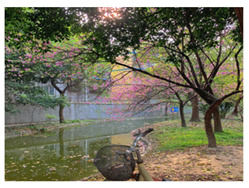	Waterscape and waterfront facilities
Gray Space	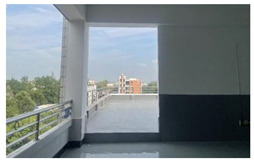	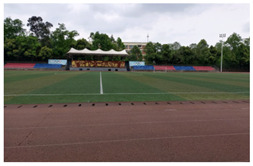	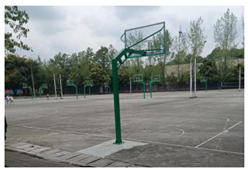	A large area of pavement
Living Space	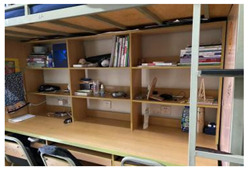	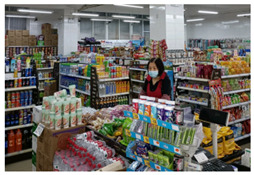	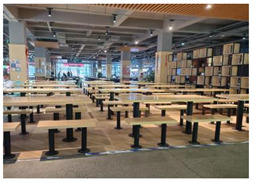	The space related to students’ daily life activities
Study Space	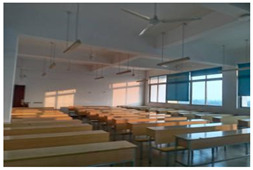	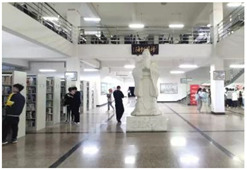	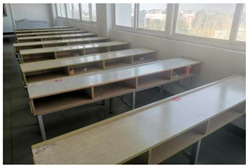	A place dedicated to learning

**Table 2 ijerph-19-13500-t002:** Restoration potential evaluation statistics for five restorative spaces.

Place	N	Mean	SD	SE	95% C.I.	
					LL	UL
Green Space	34	4.500	1.022	0.175	4.143	4.856
Blue Space	57	4.614	1.097	0.145	4.322	4.905
Gray Space	73	4.794	1.490	0.174	4.446	5.142
Living Space	48	4.729	1.440	0.207	4.311	5.147
Study Space	31	4.193	1.108	0.199	3.787	4.600
Total	243	4.621	1.293	0.083	4.457	4.784

Note: N, number of samples; SD, standard deviation; SE, standard error; C.I., confidence interval; LL, lower limit; UL, upper limit.

**Table 3 ijerph-19-13500-t003:** Analysis of the restoration scores for the five spaces (LSD).

(I) Place	(J) Place	Mean Difference (I-J)	Standard Error	Sig.	95% C.I.	
					LL	UL
Green Space	Blue Space	−0.114	0.279	1.000	−0.906	0.678
	Gray Space	−0.294	0.267	1.000	−1.053	0.464
	Living Space	−0.2291	0.289	1.000	−1.048	0.590
	Study Space	0.306	0.320	1.000	−0.601	1.214
Blue Space	Green Space	0.114	0.279	1.000	−0.678	0.906
	Gray Space	−0.180	0.228	1.000	−0.826	0.465
	Living Space	−0.115	0.252	1.000	−0.831	0.601
	Study Space	0.420	0.287	1.000	−0.395	1.236
Gray Space	Green Space	0.294	0.267	1.000	−0.464	1.053
	Blue Space	0.180	0.228	1.000	−0.465	0.826
	Living Space	0.065	0.239	1.000	−0.614	0.744
	Study Space	0.600	0.276	0.308	−0.182	1.384
Living Space	Green Space	0.229	0.289	1.000	−0.590	1.048
	Blue Space	0.115	0.252	1.000	−0.601	0.831
	Gray Space	−0.065	0.239	1.000	−0.744	0.614
	Study Space	0.535	0.297	0.729	−0.306	1.378
Study Space	Green Space	−0.306	0.320	1.000	−1.214	0.601
	Blue Space	−0.420	0.287	1.000	−1.236	0.395
	Gray Space	−0.600	0.276	0.308	−1.384	0.182
	Living Space	−0.535	0.297	0.729	−1.378	0.306

Note: C.I. = confidence interval; LL = lower limit; UL = upper limit.

**Table 4 ijerph-19-13500-t004:** Activity statistics for five restorative spaces.

	Meditation	Enjoy the View	Sports	Walk	Chat	Rest	Eat	Read	FreshAir	Nap	Total Responses	N	Activity Index
GS ^1^	11	16	3	22	14	14	5	1	18	2	106	34	3.12
BS ^2^	18	42	3	27	30	31	8	5	31	3	198	57	3.47
GrS ^3^	16	19	45	48	29	25	3	3	23	0	211	73	2.89
LS ^4^	17	2	1	6	29	30	40	7	3	13	148	48	3.08
SS ^5^	13	6	1	4	9	17	3	15	10	1	79	31	2.55
χ^2^	5.034	65.662	96.534	53.215	10.094	11.717	115.748	33.251 ^a^	32.169	26.042 ^a^			
*p*	0.284	<0.001	<0.001	<0.001	0.039	0.020	<0.001	<0.001	<0.001	<0.001			

Note: ^a^ Based on Fisher’s exact test. ^1^ GS, green space; ^2^ BS, blue space; ^3^ GrS, gray space; ^4^ LS, living space; ^5^ SS, study space.

**Table 5 ijerph-19-13500-t005:** Students’ main descriptions of restorative spaces.

Categories	Keywords	Frequency
Green Space	Here I can enjoy the natural scenery	22
	I feel comfortable and relaxed here	19
	It is quiet with few people	16
	Here I can take a walk	6
	It’s private	3
Blue Space	Here I can view the water scene	36
	It’s quiet here	32
	It’s nice and romantic here	22
	It’s fun here	8
	Here I can interact with animals	7
	Here I can take a walk	7
	I can sit around here	4
Gray Space	I can exercise here	29
	I am here for recreation	26
	It’s wide and free	22
	It’s full of energy and passion	16
	It is full of youthful sunshine	13
	Release the sweat	8
	I feel healthy here	7
Living Space	I feel lively and happy here	22
	Here I can meet the needs of life	21
	I can relax here	18
	I feel like home here	8
	I can communicate and interact here	7
	Here I can indulge in the screen	3
Study Space	There is a quiet atmosphere	21
	Here I can take a class break	19
	I can learn and read here	9
	I feel fulfilled here	6
	There is a sense of order	4

**Table 6 ijerph-19-13500-t006:** Key environmental features driving the restorative spaces.

Dependent	Independent	Unstandardized Beta	Standardized Beta	t	Sig.	Collinearity Statistics	
						Tolerance	VIF
Restoration in Green Space (adjusted R^2^ = 0.384)	(constant)	1.140		1.386	0.176		
	Being away *	0.439	0.412	2.316	0.028	0.589	1.696
	Fascination	0.311	0.241	1.413	0.168	0.644	1.553
	Coherence	0.216	0.239	1.319	0.197	0.567	1.764
	Compatibility	−0.146	−0.152	−0.941	0.355	0.711	1.407
Restoration in Blue Space (adjusted R^2^ = 0.405)	(constant)	1.302		2.488	0.016		
	Being away **	0.484	0.436	2.742	0.008	0.421	2.378
	Fascination	0.148	0.145	0.808	0.423	0.330	3.029
	Coherence	0.144	0.148	0.755	0.453	0.278	3.595
	Compatibility	0.009	0.009	0.045	0.964	0.261	3.832
Restoration in Gray Space (adjusted R^2^ = 0.504)	(constant)	−0.102		−0.172	0.864		
	Being away *	0.384	0.279	2.095	0.040	0.389	2.573
	Fascination **	0.601	0.393	3.115	0.003	0.432	2.317
	Coherence	−0.107	−0.076	−0.680	0.499	0.549	1.823
	Compatibility	0.257	0.211	1.657	0.102	0.425	2.351
Restoration in Living Space (adjusted R^2^ = 0.560)	(constant)	0.657		0.951	0.347		
	Being away **	0.479	0.393	2.813	0.007	0.479	2.086
	Fascination	0.040	0.026	0.201	0.842	0.576	1.737
	Coherence	−0.049	−0.030	−0.262	0.795	0.696	1.437
	Compatibility **	0.452	0.453	3.367	0.002	0.517	1.933
Restoration in Study Space (adjusted R^2^ = 0.298)	(constant)	1.222		1.531	0.138		
	Being away	0.027	0.025	0.127	0.900	0.627	1.595
	Fascination *	0.610	0.529	2.270	0.032	0.431	2.321
	Coherence	0.066	0.083	0.421	0.677	0.602	1.662
	Compatibility	0.043	0.046	0.180	0.859	0.355	2.821

* *p* < 0.05; ** *p* < 0.01.

## Data Availability

Not applicable.

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
