# Peer review of "Beyond Blue and Green Spaces: Identifying and Characterizing Restorative Environments on Sichuan Technology and Business University Campus"

_ijerph, 2022, doi:10.3390/ijerph192013500_

Round 1
Author Response
Response to Reviewer 1 Comments
We would like to thank you for careful and thorough reading of this manuscript and for the thoughtful comments and constructive suggestions, which help to improve the quality of this manuscript. Here is a point-by-point response to your comments and concerns. All page and line numbers refer to the revised manuscript file.
OVERALL IMPRESSION
In general, the technical structure of the manuscript meets the requirements of the Journal. An advantage of the submitted paper is the Results section. It’s well-written and presented accordingly to the research objectives presented in lines 111-116. My main concern was about the number of respondents (243 students from a Chinese university; lines 19-20). However, an explanation in lines 132-134 seems to be satisfying. You write that geographic information about the photographs was not recorded (lines 529-530). I am wondering why you didn’t want to ask respondents to provide the coordinates (GPS) when taking a photo.
I have only four suggestions:
Point 1: - Title: It should inform where the study was conducted
Response 1:
Thanks for this helpful comment. We have changed the title to "Beyond blue and green spaces: Identifying and characterizing restorative environments on Sichuan Technology and Business University campus".
Point 2: - Figure 1 – the photo (a) is illegible. I suggest adding another one with the location of Chengdu on the China map
Response 2:
Thank you for pointing this out. To make the study location clearer, we add the location of Chengdu on the China map.
Point 3: -Line 382 – delete “(2003)”
Response 3:
Thank you for pointing this out. We deleted the word of “2003”.
Point 4: - In lines 46, 50, and 55; 134 and 140; 254 and 256 there is a repetition of the word “therefore”. I suggest changing it to a synonym.
Response 4:
Thank you for pointing this out. We replaced "therefore" with “thus”(Line 53),“consequently”(Line 140),“as a result”(Line 264).
Point 5: You write that geographic information about the photographs was not recorded (lines 529-530). I am wondering why you didn’t want to ask respondents to provide the coordinates (GPS) when taking a photo.
Response 5: Thank you for pointing this out. We originally intended to collect geographic information. However, geographic distribution was not the purpose of this study, so we did not collect geographic information on the photographs during the data collection phase and hoped to focus on the research question of this study, which is the identification of potentially restorative environments on university campuses.
Additional clarifications
In addition to the above comments, all spelling and grammatical errors have been corrected. Besides, we checked the language of the manuscript again to ensure it was written correctly.
Once again, we thank you for the time you put in reviewing our paper and look forward to meeting your expectations. Since your inputs have been precious, in the eventuality of a publication, we would like to acknowledge your contribution explicitly.

Reviewer 2 Report
I appreciate that the study also included the educational environment (not only natural).
I recommend a grammar check. There are inaccuracies in the work, e.g.
243 samples was considered to be adjusted to were considered
unconsidered punctuation - you used two different styles of apostrophes or quotation marks in your document. Both styles are acceptable, but it is best to be consistent.
... with anti-depression and anti-anxiety effects that improves accumulated acute emotions [58,59] edit to improve accumulated
Congratulations on a great job and good luck
Author Response
Response to Reviewer 2 Comments
We would like to thank you for careful and thorough reading of this manuscript and for the thoughtful comments and constructive suggestions, which help to improve the quality of this manuscript. Here is a point-by-point response to your comments and concerns. All page and line numbers refer to the revised manuscript file.
OVERALL IMPRESSION
I appreciate that the study also included the educational environment (not only natural).
Point 1: - I recommend a grammar check. There are inaccuracies in the work, e.g.243 samples was considered to be adjusted to were considered.
Response 1:
Thank you for pointing this out. Corrections have been made in the manuscript. In addition, we have thoroughly checked the manuscript and avoided the use of incorrect words.
Point 2: - Unconsidered punctuation - you used two different styles of apostrophes or quotation marks in your document. Both styles are acceptable, but it is best to be consistent.
.... with anti-depression and anti-anxiety effects that improves accumulated acute emotions [58,59] edit to improve accumulated.
Response 2:
Thank you for pointing this out. We checked and modified the spelling and punctuation again.
Details are as follows:
(Line 435-438) Walking (48) and exercise (45) have been mentioned many times as activity types in gray space, and numerous studies have proven that exercise is a positive experience with anti-depression and anti-anxiety effects that improve accumulated acute emotions [61,62].
Additional clarifications
In addition to the above comments, all spelling and grammatical errors have been corrected. Besides, we checked the language of the manuscript again to ensure it was written correctly.
Once again, we thank you for the time you put in reviewing our paper and look forward to meeting your expectations. Since your inputs have been precious, in the eventuality of a publication, we would like to acknowledge your contribution explicitly.

Reviewer 3 Report
This is an interesting and novel study into the restorative effects associated with different types of environment on a university campus, assessing the benefits to students. The bottom-up approach to studying this topic, using students' smartphone photographs, has made for an interesting study, that contrasts with the top-down, researcher-driven approach of many previous studies. Your study provides useful new evidence of the health and wellbeing benefits of five different types of environment encountered by students on campus (green, blue, gray, living, and study space). The findings regarding the role of exposure duration in determining the benefits to students, is also a very interesting result that has practical applications for those working on student welfare.
Overall, the manuscript is well-written and structured clearly. It has a clearly defined set of aims presented at the end of the introduction, which give the paper a strong question-driven focus. However, I had some comments and suggestions for improvements, which I have listed below.
Specific comments:
Line 20: Please state which Chinese university featured in your study.
Lines 24-25: Can you provide brief definitions of the "being away" and "fascination" characteristics, for readers that may be unfamiliar with these terms?
Lines 37-39: "According to the latest report on national mental health development in China (2019–2020), about 18.5% of college students suffer from depression, and their mental health problems are increasing." The report mentioned in this statement should be cited.
Lines 39-41: "According to the American College Health Association, overall anxiety among college students has increased by 14.2% since the year 2000." A citation of the relevant paper/report should be provided here.
Lines 45-46: "Because of the travel restrictions set in place during the COVID-19 pandemic, students are spending even more time on campus". This point was also made in this previous relevant study of the effects of different on-campus environments on student wellbeing, which could be cited here: Sun, S., et al. (2021). The psychological restorative effects of campus environments on college students in the context of the COVID-19 pandemic: a case study at Northwest A&F University, Shaanxi, China. International Journal of Environmental Research and Public Health, 18 (16), 8731.
Lines 55-57: It would be informative to cite some of these studies here, for example: Seitz, C.M., et al. (2014). Identifying and improving green spaces on a college campus: A photovoice study. Ecopsychology, 6(2), 98-108.
Line 62: The previous sentences offer examples (e.g. tree-lined walkways) of green spaces, but no examples are given for blue spaces, and so there is a risk that the reader may not understand what is meant by "blue spaces" or why they might be beneficial to health and well-being. You should add text here to briefly describe what is meant by blue spaces, giving examples such as wetlands, ponds, lakes, and rivers. There are examples of blue spaces and their health benefits from previous papers in this journal that can be cited in support of this new statement, for example: Reeves, J.P., et al. (2021). A qualitative analysis of UK wetland visitor centres as a health resource. International Journal of Environmental Research and Public Health, 18 (16), 8629.
Line 119: Please state which Chinese university featured in your study.
Lines 152-153: Over what time period should these visits have occurred? Were you interested in the number of visits per year, or over the entire duration of the student's time at the university?
Lines 199-202: What do the values reported here represent? What units are these values measured in? Furthermore, are they mean +/- standard deviation, or something else?
Line 293: A label should be added for the y-axis on Figure 2.
Lines 535-536: "While blue and green spaces have already been proven to be restorative by previous research". This statement should be supported by appropriate references to research on the restorative benefits of blue and green spaces, such as the references that I mentioned in my comments on lines 55 and 62.
Author Response
Response to Reviewer 3 Comments
We would like to thank you for careful and thorough reading of this manuscript and for the thoughtful comments and constructive suggestions, which help to improve the quality of this manuscript. Here is a point-by-point response to your comments and concerns. All page and line numbers refer to the revised manuscript file.
OVERALL IMPRESSION
This is an interesting and novel study into the restorative effects associated with different types of environment on a university campus, assessing the benefits to students. The bottom-up approach to studying this topic, using students' smartphone photographs, has made for an interesting study, that contrasts with the top-down, researcher-driven approach of many previous studies. Your study provides useful new evidence of the health and wellbeing benefits of five different types of environment encountered by students on campus (green, blue, gray, living, and study space). The findings regarding the role of exposure duration in determining the benefits to students, is also a very interesting result that has practical applications for those working on student welfare.
Overall, the manuscript is well-written and structured clearly. It has a clearly defined set of aims presented at the end of the introduction, which give the paper a strong question-driven focus. However, I had some comments and suggestions for improvements, which I have listed below.
Point 1: - Line 20: Please state which Chinese university featured in your study.
Response 1:
Thank you for pointing this out. We added the name of the university to the manuscript.
Details are as follows:
(Line 20-22) Using a bottom-up participatory smartphone photo survey, this study recruited a sample of 243 students from Sichuan Technology and Business University in China, and the results are as follows:
Point 2: - Lines 24-25: Can you provide brief definitions of the "being away" and "fascination" characteristics, for readers that may be unfamiliar with these terms?
Response 2:
Thank you for pointing this out. We added a brief definition to manuscript.
Details are as follows:
(Line 24-28) the five restorative spaces were linked with four restorative characteristics in different ways, with green, blue, gray, and living space showing the “being away” characteristic (refuges from the hassles of everyday life, indicate geographical or psychological distance), and gray and study spaces showing the “fascination” characteristic (effortless attention);
Point 3: -Lines 37-39: "According to the latest report on national mental health development in China (2019–2020), about 18.5% of college students suffer from depression, and their mental health problems are increasing." The report mentioned in this statement should be cited.
Response 3:
Thank you for pointing this out. We added the reference to this report in the manuscript.
Point 4:-Lines 43-45: "According to the American College Health Association, overall anxiety among college students has increased by 14.2% since 2000 [6]." A citation of the relevant paper/report should be provided here.
Response 4:
Thank you for pointing this out. We added the reference to this report in the manuscript.
Point 5:-Lines 45-46: "Because of the travel restrictions set in place during the COVID-19 pandemic, students are spending even more time on campus". This point was also made in this previous relevant study of the effects of different on-campus environments on student wellbeing, which could be cited here: Sun, S., et al. (2021). The psychological restorative effects of campus environments on college students in the context of the COVID-19 pandemic: a case study at Northwest A&F University, Shaanxi, China. International Journal of Environmental Research and Public Health, 18 (16), 8731.
Response 5:
Thank you for providing this helpful reference. We have added this reference to our manuscript.
Details are as follows:
(Line 48-50) Because of the travel restrictions set in place during the COVID-19 pandemic, students spend even more time on campus. Therefore, it is necessary and urgent to investigate restorative sites on campus [8].
Point 6:-Lines 55-57: It would be informative to cite some of these studies here, for example: Seitz, C.M., et al. (2014). Identifying and improving green spaces on a college campus: A photovoice study. Ecopsychology, 6(2), 98-108.
Response 6:
Thank you for providing this helpful reference. We have added this reference to our manuscript.
Details are as follows:
(Line 61-65) …and the green space gives students the opportunity to easily immerse themselves in soft fascination with nature during recess, freeing them from the stress of work and study [15]. Green spaces have the ability of helping students relax [16].
Point 7:-Line 62: The previous sentences offer examples (e.g. tree-lined walkways) of green spaces, but no examples are given for blue spaces, and so there is a risk that the reader may not understand what is meant by "blue spaces" or why they might be beneficial to health and well-being. You should add text here to briefly describe what is meant by blue spaces, giving examples such as wetlands, ponds, lakes, and rivers. There are examples of blue spaces and their health benefits from previous papers in this journal that can be cited in support of this new statement, for example: Reeves, J.P., et al. (2021). A qualitative analysis of UK wetland visitor centres as a health resource. International Journal of Environmental Research and Public Health, 18 (16), 8629.
Response 7:
Thank you for pointing this out. We have added a short description of the blue space here. Besides, we added this reference to our manuscript.
Details are as follows:
(Line 66-69) Blue spaces (a natural environment dominated by water bodies, which correspond to green spaces, e.g., wetlands, rivers, lakes, and ponds) are considered to have the best restoration potential among outdoor campus spaces [8,18].
Point 8:-Line 119: Please state which Chinese university featured in your study.
Response 8:
Thank you for pointing this out. We added the name of the university we studied at line 125.
Details are as follows:
(Line 125-126) Sichuan Technology and Business University of China was selected as the study area.
Point 9:-Lines 152-153: Over what time period should these visits have occurred? Were you interested in the number of visits per year, or over the entire duration of the student's time at the university?
Response 9:
Thank you for this comment.
We are very sorry that we forgot to add the period of the experiment. In the revised version, we have added the period of the experiment.
Details are as follows:
(Line 149-151) Participants were asked to take a photo inside the school and complete a questionnaire based on it, limited to the daytime for clearer photos. The photographic survey was conducted from March 20 to April 1. .
Besides, the frequency of visits was asked as "so far this semester" for students to clearly recall the frequency of visits. We have added this information to the manuscript.
Details are as follows:
(Line 158-162) Utilization patterns were divided into two questions: usage duration (‘What is the average time spent here per visit?’ where 1= less than 30 min, 2= 30-60 min, 3= 1-2 hours, 4= more than 2 hours), and frequency of visits (“How often have you visited the site so far this semester?” where 1= Hardly visited, 2= Occasionally, 3= Frequently, 4= Almost daily).
Point 10:-Lines 199-202: What do the values reported here represent? What units are these values measured in? Furthermore, are they mean +/- standard deviation, or something else?
Response 10:
Thanks for this comment. The values reported here indicate the student's assessment of the restoration potential of the spaces shown in the photographs. The description of this value measurement is in lines 163 to 167 of the manuscript. Besides, as mentioned in this comment, the values inside the parentheses represent mean +/- standard deviation, which is not explained in detail in the manuscript since it is a common representation.
Details are as follows:
(Line 163-167) First, as Jiang asserted [23], we asked participants for their perceived restorative assessment of these sites using a simple question: ‘Based on your own actual visit experience, to what extent do you think this place meets your needs for stress relief and spiritual recovery?’ On a seven-point Likert scale, 1 means not at all and 7 means extremely good.
Point 11:-Line 293: A label should be added for the y-axis on Figure 2.
Response 11:
Thanks for point this out. We added a simple label for the y-axis on Figure 2. However, since the values represented by the y-axis are only the participants' responses to duration and frequency, there are no units. Please refer to lines 158-162 for the corresponding description of duration and frequency.
Details are as follows:
(Line 158-162) Utilization patterns were divided into two questions: usage duration (‘What is the average time spent here per visit?’ where 1= less than 30 min, 2= 30-60 min, 3= 1-2 hours, 4= more than 2 hours), and frequency of visits (“How often have you visited the site so far this semester?” where 1= Hardly visited, 2= Occasionally, 3= Frequently, 4= Almost daily).
Point 12:-Lines 535-536: "While blue and green spaces have already been proven to be restorative by previous research". This statement should be supported by appropriate references to research on the restorative benefits of blue and green spaces, such as the references that I mentioned in my comments on lines 55 and 62.
Response 12:
Thank you for pointing this out. We added relevant references to our manuscript.
Details are as follows:
(Line 546-548) Although blue and green spaces have already been proven to be restorative in previous research [8,14-18], the results of this study indicate that gray, living, and study spaces on campus are also restorative.
Additional clarifications
In addition to the above comments, all spelling and grammatical errors have been corrected. Besides, we checked the language of the manuscript again to ensure it was written correctly.
Once again, we thank you for the time you put in reviewing our paper and look forward to meeting your expectations. Since your inputs have been precious, in the eventuality of a publication, we would like to acknowledge your contribution explicitly.
